# Potentially Toxic Element Availability and Risk Assessment of Cadmium Dietary Exposure after Repeated Croppings of *Brassica juncea* in a Contaminated Agricultural Soil

**Diana Agrelli [1], Luigi Giuseppe Duri [2,\*], Nunzio Fiorentino [1,2], Eugenio Cozzolino [3], Massimo Fagnano [1,2] and Paola Adamo [1,2]**

[1] CIRAM-Interdepartmental Center for Environmental Research, University of Naples Federico II, Via Mezzocannone 16, 80134 Naples, Italy; d.agrelli@unina.it (D.A.); nunzio.fiorentino@unina.it (N.F.); fagnano@unina.it (M.F.); paola.adamo@unina.it (P.A.)

[2] Department of Agricultural Sciences, University of Naples Federico II, via Università 100, 80055 Portici, Naples, Italy

[3] Council for Agricultural Research and Economics (CREA)-Research Center for Cereal and Industrial Crops, via Torrino 2, 81100 Caserta, Italy; eugenio.cozzolino@crea.gov.it

\* Correspondence: lgduri@libero.it

**Abstract:** Phytoextraction of potentially toxic elements (PTEs) is eco-friendly and cost-effective for remediating agricultural contaminated soils, but plants can only take up bioavailable forms of PTEs, thus meaning that bioavailability is the key for the feasibility of this technique. With the aims to assess the phytoextraction efficiency on an agricultural soil contaminated by Cr, Zn, Cd, and Pb and the changes induced by plants in PTE bioavailability and in human health risk due to dietary exposure, in this work we carried out a mesocosm experiment with three successive croppings of *Brassica juncea*, each followed by Rocket salad as bioindicator. *Brassica juncea* extracted more Zn and Cd than Cr and Pb, significantly reducing, after three repeated croppings, the bioavailable element concentrations in soil as a result of plant uptake and soil pH changes. For Cd, this reduction did not bring the bioavailable amounts obtained by soil extraction with $NH_4NO_3$ below the trigger value of 0.1 mg kg$^{-1}$ set by some European countries. Nevertheless, the Hazard Quotient for Cd in Rocket salad decreased across three repeated croppings of *Brassica juncea*. This indicated the beginning of a re-equilibration process between soil PTE forms of different bioavailability, that are in a dynamic equilibrium, thus stressing the need to monitor the possible regeneration of the most readily bioavailable pool.

**Keywords:** bioavailability; ammonium nitrate; EDTA; Rocket salad; phytoextraction efficiency; Hazard Quotient; reclamation time

## 1. Introduction

Soil contamination by potentially toxic elements (PTEs) is of great concern due to their harmfulness for the biota at certain concentrations and to their persistence in the environment [1]. When contaminant concentrations in a site overcome the screening values stated by the national legislations, the site has to be subjected to a site-specific risk analysis for calculating risk thresholds. When contaminant concentrations overcome also risk thresholds, the site must be subjected to remediation projects [2]. In the case of agricultural soils, it is crucial to also analyze the risks that a contaminant can enter the food chain, for this purpose chemical [3] and biological essays [4] have been proposed.

Various soil remediation techniques exist, but many of them are complex and expensive. In recent years, much research has been done on soil phytoremediation because it is an in situ eco-friendly and

cost-effective method with respect to the others [5]. The main phytoremediation mechanisms are the phytostabilization, which reduces contaminant mobility toward other environmental compartments, and the phytoextraction, which exploits the ability of some plants to take up potentially toxic elements from the contaminated soil [6]. The best suitable plants for phytoextraction are the hyperaccumulators, i.e., plants capable of growing in soils with very high concentrations of PTEs they accumulate in extremely high levels in their tissues [7]. As an alternative, fast-growing high biomass crops that accumulate moderate levels of PTEs in their shoots can allow a greater phytoextraction efficacy since the high biomass yield can more than compensate for the lower PTE concentration in plant tissues [8]. In any case, the contaminant to be absorbed by the plants must be in a bioavailable form in the soil [1]. PTEs in the soil from anthropogenic sources tend to be more mobile and bioavailable than the geogenic ones [5]. This may promote their transfer to other environmental compartments and to the food chain. Therefore, a realistic remediation objective through phytoextraction can be the progressive reduction of the contaminant to safety levels of its bioavailable portion rather than its total removal [9]. However, the bioavailability of a PTE is closely linked to the nature of the element, to the chemical forms in which it occurs in soil, and to the chemical and physical characteristics of the soil itself. This determines the repartition of the element between the various soil geochemical fractions and the soil solution (in which the element is in a readily bioavailable form) [3,10]. This repartition is controlled by dynamic equilibria between different forms elements and soil fractions. It must be considered in planning strategies of phytoremediation. After short-term phytoextraction (or only few cycles of phytoextraction), the readily bioavailable fraction of the respective element may be replenished through repartition between soil fractions and soil solution, and the kinetics of replenishment can change over time [9].

In view of these considerations, in the present work we report the results of a 3-year mesocosm experiment of PTEs (mainly cadmium) phytoextraction from an agricultural soil polluted by illegal dumping of industrial wastes and by consecutive croppings of *Brassica juncea* L. (common name Indian mustard). At the end of each cropping cycle, chemical extractions and a bioassay with *Eruca vesicaria* L. (common name Rocket salad) were done to assess the bioavailability of the residual PTEs and thus the opportunity to reutilize the treated soil for food production. Specific work aims were: (i) to study the PTEs repartition between total, potentially bioavailable and readily bioavailable pools in soil; (ii) to follow the growth of *Brassica juncea* L. on polluted soil and the plant PTEs uptake; (iii) to evaluate the changes in soil properties and PTEs repartition that may occur during the phytoextraction cycles by a combined approach of selective chemical extractions and bioindicator plant cultivation as the rocket salad; (iv) to assess the risk of dietary exposure to cadmium accumulated in vegetables cultivated on the polluted soil.

Based on the results, two main hypotheses were tested: (i) to evaluate the possibility to phytoclean the soil and to estimate the time-span of soil phytoremediation; (ii) to assess the feasibility of an agricultural use of the soil after remediation.

## 2. Materials and Methods

### 2.1. The Study Site

The study site is an agricultural land of 6 ha, currently under sequestration, located in the province of Naples (Campania, south Italy), contaminated by former illegal dumping of industrial wastes, mainly from tanneries. The soil of the area was highly enriched by several potentially toxic elements (PTEs) such as Cr (up to 4500 mg kg$^{-1}$), Zn (up to 1850 mg kg$^{-1}$), Cd (up to 280 mg kg$^{-1}$), and Pb (up to 420 mg kg$^{-1}$) [11]. For soil properties and PTE total content, a high spatial variability, both horizontal and vertical, was observed [11,12]. Chromium and Zn were found in almost all regions. In contrast, Cd and Pb were localized in few sub-areas. According to sequential and selective chemical extractions, only the mobility and availability of cadmium could represent a risk for food chain contamination. The protocol of phytoremediation implemented in the ECOREMED project (LIFE11/ENV/IT/275) and based on the use of poplar trees and grass species was applied for securing the site [6].

With the soil from the sub-areas where Cd and Pb were detected, a phytoremediation mesocosm open air experiment was set up with the idea to intensify the phytoextraction process by consecutive *Brassica juncea* L. cropping cycles. Although we were aware of the fact that plant growth out in field and in mesocosms can be different, we decided to set up the mesocosm experiment in order to reduce the high spatial variability of field contamination, which depreciates comparisons between results obtained at different times.

### 2.2. The Mesocosm Experiment

Soil excavation (0–30 cm) and sampling from three field sub-areas (F4, C13, A7) were carried out for the mesocosm experiment. Soils F4 and C13 were identified as Cd and Pb contaminated, respectively, on the basis of total contents. Soil A7 was collected from a sub-area in which no legal thresholds of PTEs were surpassed, and it was considered as control. After each excavation, the soils were homogenized and transported to the university facilities. Four plastic mesocosms with 7 kg fresh weight of each homogenized soil were prepared. Three crops of Indian mustard (*Brassica juncea* L. Czern.) followed by Rocket salad (*Eruca vesicaria* L. Cav.) were established over 3 consecutive years. Fifteen seeds of Indian mustard (Semfor S.r.l., San Pietro Di Morubio, Italy) were sown in each mesocosm (in five points following an X shape) and a few weeks later the five more robust plants per mesocosm were selected and the others removed. At the end of each cycle (end of flowering and beginning seed maturation), the five plants per mesocosm were harvested and divided in roots, stems and leaves, weighed, washed with tap water for removing soil particles, and successively with deionized water. For each sample, a portion was dried at 60 °C to constant weight to determine the dry matter and the rest was used to determine the PTEs total content. During harvest, the rhizosphere soil adhering to the roots was also sampled from each mesocosm for the chemical analyses. After each cycle of phytoextraction with Indian mustard crop, in each mesocosm, Rocket salad was sown as a food-plant bioindicator of the PTEs bioavailability. The Rocket salad was cultivated simulating a normal productive process. At commercial maturity, the leaves were cut four times and the PTE content was only analyzed in the edible parts at the end of the first and fourth cutting.

Indian mustard growth periods were: 15 October 2016 to 24 March 2017 (160 d); 24 November 2017 to 13 April 2018 (140 d); 13 November 2018 to 16 April 2019 (154 d). Rocket salad growth periods from sowing to the 4th harvest were: 3 April to 7 July 2017 (95 d); 24 April to 1 August 2018 (99 d); and 6 May to 8 August 2019 (94 d).

All mesocosms (D = 33 cm; V = 7 dm$^3$) equipped with water reservoirs for avoiding water deficits were set in open air at the experimental fields of the Department of Agricultural Sciences, University of Naples Federico II, southern Italy (40°49′ N, 14°15′ E).

About the fertilizer, a preliminary physicochemical analysis showed a high content of phosphorus, by 59 up to 240 mg kg$^{-1}$ of $P_2O_5$ (using Olsen method to analyzing); and a very high content of potassium, by 2715 up to 6002 mg kg$^{-1}$ of $K_2O$ (through tetraphenylborate colorimetric method); for this reason, only nitrogen was distributed (using ammonium nitrate 26% N). The fertilizer dose for *Brassica juncea* was 1.97 g mesocosm$^{-1}$, distributed twice, the first-half after the thinning out and the second half at the grow-up phase. While the fertilizer dose for *Eruca vesicaria* was given four times, the first after the sowing (1.30 g mesocosm$^{-1}$) and the other three after each harvest (0.49 g mesocosm$^{-1}$).

It is worth mentioning a late and intense snowfall in the 1st week of March 2018, which is unusual in plain areas of Mediterranean regions, that injured Indian mustard plants during flowering and slowed their growth.

### 2.3. Soil and Plant Analysis

At the beginning of the mesocosm experiment and after each of the three cycles of Indian mustard, the soil in each mesocosm (3 soils × 4 replicates) was sampled and analyzed for pH, organic carbon

content, and pseudo-total and bioavailable content of the potentially toxic elements (PTEs) of interest (Cd, Cr, Pb, Zn).

The soils were air-dried and sieved to 2 mm, then the pH was determined potentiometrically by applying a 1:2.5 soil: water ratio, and organic carbon was determined according to the Walkley and Black method [13].

The determination of pseudo-total content of the PTEs of interest was carried out on dried and pulverized soils by microwave-assisted aqua regia digestion and ICP-MS analysis of the digestion solutions [14,15].

The determination of the readily and potentially bioavailable amounts of the PTEs of interest in soil was carried out respectively on dried and 2 mm sieved soils by extractions in $NH_4NO_3$ 1 M (2 h of extraction in a ratio soil: extractant of 1:2.5; [16]) and in ethylenediaminetetraacetic acid (EDTA) 0.05 M pH 7 (1 h of extraction in a ratio soil: extractant of 1:10; EUR 19774 EN/2001 from Rauret et al. [17]). The extraction solutions were then analyzed by atomic absorption spectrometer (AAnalyst 700, Perkin Elmer, Norwialk, CT, USA).

Plant analysis (roots, stems and leaves of Indian mustard and leaves of Rocket salad) was carried out by microwave-assisted nitric acid digestion and ICP-MS PTEs determination in digestion solutions.

## 2.4. Risk Assessment of Dietary Exposure

The hazard quotient (HQ) method [18] was performed to evaluate the potential non-cancer risk, following the calculation formula as explained by Duri et al. [4].

HQ is calculated as the ratio between the average daily dose (ADD) of a PTE ingested and its reference dose (RfD). The values necessary for calculating the daily ingestion of PTEs (Table S1) were found using national and international databases [19,20], while the PTEs content was given by fresh leaves analysis.

When ADD is equal or higher than RfD (i.e., HQ ≥ 1), there is a potential risk for human health due to dietary exposure.

## 2.5. Statistical Analysis

Correlation analysis and analysis of variance (ANOVA) followed by Fischer LSD test for multiple comparisons on soils data were carried out by XLstat 2009.3.02 software (Excel add-in, Microsoft Corporation, Redmond, WA, USA); differences were considered significant for $p < 0.05$.

## 3. Results and Discussion

### 3.1. Soil Properties and PTEs Bioavailability Assessed by Chemical Extractions before Plant Growth

The soil of the whole study site was fully characterized for its physical and chemical properties and results are given in Agrelli et al. [11]. Briefly, the soil of the study site showed sandy texture (ISSS classification) and a certain variability of the other properties as function of the contamination level. The three sub-areas selected for this work showed neutral pH (7.1 ± 0.1 in A7, 7.0 ± 0.1 in C13 and 7.1 ± 0.1 in F4) and variable organic carbon content (13.1 ± 0.2, 25.9 ± 0.3 and 16.9 ± 0.2 g kg$^{-1}$ in A7, C13 and F4 respectively).

The pseudo-total (aqua regia digestion), potentially bioavailable (EDTA extractable) and readily bioavailable ($NH_4NO_3$ extractable) contents of Cr, Zn, Cd, and Pb in the soils collected from A7, C13 and F4 sub-areas are shown in Table 1.

Total contents of Cr, Zn and Cd in F4 soil and of Cr, Zn and Pb in C13 soil exceed the thresholds set by Italian D.Lgs 46/2019 for agricultural soils (150, 300, 5 and 100 mg kg$^{-1}$ for Cr, Zn, Cd and Pb respectively). In F4 soil, Cr and Cd were almost 5-fold and 2.5-fold higher than the legal thresholds; Zn was just a little above the threshold. In C13 soil, Cr, Zn and Pb were 4-fold and 2.5-fold higher than the respective thresholds. As expected, all elements occurring in soil from A7 were in amounts below

the legal thresholds. This means that both F4 and C13 soils can be considered potentially contaminated and need further investigation on the risk of contaminants transfer to humans and food chain.

**Table 1.** Total, potentially bioavailable (EDTA) and readily bioavailable (NH$_4$NO$_3$) concentrations (mg kg$^{-1}$ ± standard deviation) of Cr, Zn, Cd, and Pb in the soils from A7, C13 and F4 sub-areas.

|  |  | **Cr** | **Zn** | **Cd** | **Pb** |
|---|---|---|---|---|---|
| **A7** | Total | 112 ± 2 | 122 ± 2 | 0.19 ± 0.02 | 45 ± 1 |
|  | EDTA | 1.4 ± 0.1 | 31.0 ± 0.9 | 0.22 ± 0.02 | 10.7 ± 0.1 |
|  | NH$_4$NO$_3$ | 0.30 ± 0.01 | 0.14 ± 0.01 | 0.049 ± 0.002 | <0.1 |
| **C13** | Total | 650 ± 6 | 684 ± 7 | 0.47 ± 0.02 | 147 ± 2 |
|  | EDTA | 6.2 ± 0.1 | 349 ± 1 | 0.52 ± 0.03 | 73.8 ± 0.1 |
|  | NH$_4$NO$_3$ | 0.54 ± 0.01 | 0.82 ± 0.01 | 0.068 ± 0.002 | <0.1 |
| **F4** | Total | 716 ± 8 | 342 ± 3 | 13.5 ± 0.1 | 70 ± 1 |
|  | EDTA | 2.5 ± 0.1 | 127 ± 0 | 8.46 ± 0.01 | 20.4 ± 1.4 |
|  | NH$_4$NO$_3$ | 0.50 ± 0.00 | 0.49 ± 0.02 | 0.18 ± 0.01 | <0.1 |

Average amounts extracted by EDTA ranged from 1.4 (A7) to 6.2 (C13) mg kg$^{-1}$ for Cr (about or lower than 1% of the total), from 31 (A7) to 349 (C13) mg kg$^{-1}$ for Zn (25–51% of the total), from 0.2 (A7) to 8.5 (F4) mg kg$^{-1}$ for Cd (62 to 100% of the total), and from 11 (A7) to 74 (C13) mg kg$^{-1}$ for Pb (24–50% of the total). A positive relation between EDTA extractable amounts and total content was observed with the lower EDTA extractable amounts always associated to the non-potentially contaminated A7 soil.

Average amounts extracted by NH$_4$NO$_3$ were below 1 mg kg$^{-1}$ for all elements in all soils. NH$_4$NO$_3$ extractable amounts correspond to <1% of the total for Cr, Zn and Pb, and to 1–26% of the total for Cd. Also the NH$_4$NO$_3$ extractable amounts were positively related to totals and again the lowest values were measured in the A7 soil. Only for Cd in F4 soil, the NH$_4$NO$_3$ extractable amounts exceeded the trigger value of 0.1 mg kg$^{-1}$ adopted by some European countries as action values for the pollutant transition soil-food plant on agricultural areas and in vegetable gardens with regard to the plant quality [21].

The soil pH and organic carbon are very important soil properties that govern elements solubility, influencing their adsorption/desorption equilibrium [22–25]. It is reported that the investigated elements decrease their solubility at alkaline pH [23]; therefore, the neutral pH of the three studied soils did not constitute an obstacle to their solubilization. The different PTE amounts extracted in EDTA and NH$_4$NO$_3$ in the three soils can be related primarily to the different soil total contents and to the different intrinsic hydrolytic properties of the various elements due to their ability to form hydroxyl complexes [26]. Indeed, cadmium, with the highest pK value [26], was extracted in higher percentage than chromium, which, despite the high contents in C13 and F4 soils, was extracted in very low amounts due to its high ability to strongly bind a variety of ligands in soil [27]. In addition, neutral pH and soil organic matter do not favor the stabilization of the Cr (VI) (more mobile Cr form) in this soils, and it is conceivable that almost all the chromium was in the trivalent immobile form [11]. C13 was the soil with the highest potentially bioavailable pool of Cr, Zn and Pb, and probably the high organic carbon content constitutes the reserve for these elements [28].

In general, the data highlight the very high extractability of cadmium in respect to the other elements. This thing makes cadmium dangerous for the transfer to the plants and food chain, but also phytoextractable. Chromium was almost immobile. Zinc and lead have a high potentially bioavailable pool in the soil contaminated by them, but lead was more scarcely mobilized than zinc toward the readily bioavailable pool. This was testified by the EDTA/NH$_4$NO$_3$ ratio that for Pb was >700 in C13 soil, the only soil contaminated by Pb, and for Zn was approximatively between 200 and 400. The ratio between potentially (EDTA-extractable) and readily (NH$_4$NO$_3$-extractable) bioavailable amounts might be taken as an indicator of the mobility of the studied PTEs in soil and of the capacity of the soil

to replenish the readily bioavailable content to reach an equilibrium condition between these two pools. A higher ratio can indicate the poor mobility or tendency of PTEs to switch from the potentially bioavailable pool to the readily bioavailable pool. In the studied soils, Cd had a low ratio of about 46 in F4 and about 5 and 8 in A7 and C13 soils, confirming the great potential mobility of this element. The exception to this interpretation was for chromium that showed the lower ratio (between 5 and 11), but this element was extracted in very low amount also in the potentially bioavailable pool due to its intrinsic immobility at neutral-subalkaline soil pH, and this distorts the interpretation of the ratio.

### 3.2. PTEs Phytoextraction by Indian Mustard

Biomass and elemental composition of Indian mustard are presented in Table 2. The corresponding analysis of variance is given in Table S2. Overall and yearly biomass production was significantly larger in soil C13 than in soils F4 and A7. No visible phytotoxic symptoms—such as narrow yellowish leaves, with small necrotic spots [29], or decreases in seed germination, plant growth and biomass yield [30]—were recorded on Indian mustard plants across the three croppings in all soils. However, the biomass was always significantly lower for the second cropping, due to plant injuries for a late snowfall recorded in 2018 growing season.

**Table 2.** Biomass production expressed in g dry weight (d.w.) mesocosm$^{-1}$ ± standard deviation and selected element concentrations (mg PTE kg$^{-1}$ d.w. ± standard deviation) in Indian mustard plants for the three successive croppings on the three studied soils (different letters indicate significant differences between the croppings in a given soil, for $p < 0.05$).

| Soils | Years | Weight (g) | | Cr (mg kg$^{-1}$) | | Zn (mg kg$^{-1}$) | | Cd (mg kg$^{-1}$) | | Pb (mg kg$^{-1}$) | |
|---|---|---|---|---|---|---|---|---|---|---|---|
| A7 | 2017 | 198.2 | ±9.4 e | 2.2 | ±0.1 b | 123.6 | ±7.4 b | 0.3 | ±0.0 c | 0.7 | ±0.1 |
| | 2018 | 106.7 | ±5.4 g | 2.3 | ±0.2 b | 52.5 | ±3.4 c | 0.1 | ±0.0 c | 0.7 | ±0.1 |
| | 2019 | 231.0 | ±8.9 c | 3.0 | ±0.1 b | 22.1 | ±1.2 c | 0.1 | ±0.0 c | 0.7 | ±0.1 |
| | *Avg.*[*] | *178.6* | | *2.5* | | *66.1* | | *0.2* | | *0.7* | b |
| F4 | 2017 | 207.7 | ±9.5 e | 2.9 | ±0.1 b | 137.7 | ±7.7 a | 8.2 | ±0.3 a | 0.8 | ±0.2 |
| | 2018 | 96.6 | ±7.7 g | 4.2 | ±0.5 b | 78.0 | ±4.1 c | 4.7 | ±0.1 b | 0.7 | ±0.1 |
| | 2019 | 268.9 | ±5.2 a | 6.2 | ±0.3 b | 29.1 | ±0.9 c | 4.3 | ±0.3 b | 0.8 | ±0.0 |
| | *Avg.*[*] | *191.1* | | *4.4* | | *81.6* | | *5.7* | | *0.8* | b |
| C13 | 2017 | 218.2 | ±5.4 d | 4.0 | ±0.3 b | 136.4 | ±7.6 a | 0.1 | ±0.0 c | 0.8 | ±0.0 |
| | 2018 | 126.5 | ±3.8 f | 3.3 | ±0.1 b | 113.4 | ±3.4 b | 0.1 | ±0.0 c | 1.8 | ±0.6 |
| | 2019 | 236.5 | ±20.3 b | 7.7 | ±0.8 a | 40.8 | ±2.2 c | 0.1 | ±0.0 c | 1.5 | ±0.2 |
| | *Avg.*[*] | *193.7* | | *5.0* | | *96.9* | | *0.1* | | *1.4* | a |
| Average | 2017 | *208.0* | | *3.0* | | *132.6* | | *2.9* | | *0.7* | |
| | 2018 | *109.9* | | *3.3* | | *81.3* | | *1.6* | | *1.1* | |
| | 2019 | *245.5* | | *5.6* | | *30.7* | | *1.5* | | *1.0* | |
| | *Avg.*[*] | *187.8* | | *4.0* | | *81.5* | | *2.0* | | *0.9* | |

[*] Avg. = Average of three years.

On average, Cr and Pb concentrations were up to two times higher in Indian mustard plants grown in C13 soil with a significant increasing trend with time only for Cr. Zinc concentrations were above 100 mg kg$^{-1}$ in first cropping plants (in all soils), and in second cropping plants (only in C13) with a decreasing trend across the three croppings. Cadmium concentrations were significantly above 0.1 mg kg$^{-1}$ only in Indian mustard grown in soil F4 where values ranged between 8.2 and 4.3 mg kg$^{-1}$ passing from the first to the third cropping.

The decreasing trend of Cd and Zn concentrations in *Brassica juncea* plants across the three croppings is well in agreement with the observed overall reduction of the readily bioavailable contents (as assessed by NH$_4$NO$_3$ extractions) of these elements after the second and third growing cycles (see Section 3.3. Changes in soil properties and PTE bioavailability). The high solubility of Cd and Zn in soil at neutral and subalkaline pH (higher than that of Cr and Pb) along with the high phytoextraction capacity of Cd and Zn by Indian mustard [31] might have produced a reduction of the soil readily

bioavailable pool of these elements across the three repeated croppings. However, the corresponding changes of the potentially bioavailable Cd and Zn (increasing and decreasing respectively across the three croppings) might be the early sign of soil re-equilibration. In this sense, some authors stress that the phytoextraction efficiency must be assessed by different approaches in order not to overlook any potential hazard and that an efficient phytoextraction scheme will have to take into account the different dynamics of the soil-plant system [9].

The higher mobility of these two PTEs was also suggested by the bioconcentration factors (i.e., the ratio of the PTE concentration in the plant shoots to the concentration of the same PTE in the soil), that at the beginning of the experiment were higher for Cd and Zn (0.58 and 0.59 respectively) than for Pb and Cr (0.01), thus confirming the findings of Visconti et al. [32] who measured higher Cd and Zn bioconcentration coefficients and translocation factors from roots to shoots in many species. Also, the mobility from roots to shoots (i.e., the translocation factor) was much higher for Zn (2.9) and Cd (1.6), which suggests the suitability of Indian mustard for phytoextraction of these PTEs [32]. Otherwise, also the translocation factor was very low for Cr (0.4) and Pb (0.2), thus confirming the low mobility of these PTEs in these conditions and their preferential accumulation in roots [33,34].

The low values of Cr concentration in Indian mustard plants, other than to the element's low mobility due to the neutral-subalkaline soil pH, could also be related to the high soil organic matter content of C13 and F4 [35]. The increase in Cr concentration in plants, after 3 years of Indian mustard cultivation in all soils, could be related to the decrease in soil organic matter observed in soils at the end of the experiment (see Section 3.3. Changes in soil properties).

In Figure 1, the PTE concentration in plant tissues of *Brassica juncea* grown in mesocosms on the studied soils are given as average values of the three repeated croppings. The corresponding analysis of variance is given in Table S4. The interaction soils × plant organs were significant for Cr and Cd, while for Zn and Pb only the differences among the plant organs resulted in significance (Table S4). Chromium was preferentially accumulated in the roots where concentrations were significantly higher than in the other plant tissues only in the potentially contaminated soils C13 and F4. A similar behaviour was observed for Pb concentrations which were significantly higher in roots than in shoots and leaves in all soils. The higher concentration of Cr and Pb in plant roots again indicates an important restriction of the internal transport of these metals from the roots towards stems and leaves. In contrast, Zn was preferentially accumulated in leaves where concentrations were higher than in shoots and roots. Similar distribution was observed for Cd, which in F4 soil was accumulated in leaves in amounts significantly higher than in shoots and roots, thus confirming the well-known capacity of Indian mustard to accumulate Zn and Cd in shoots [36].

In addition, a correlation analysis (Table S5) between the PTEs concentration in the whole plant with the concentration measured in the soil as total, potentially bioavailable (EDTA) and ready bioavailable fractions ($NH_4NO_3$), showed the absence of correlations for chromium ($r < 0.58$ for all plant and extractable fractions), thus confirming the very low bioavailability acceptance by plants for this element [28]. It accumulated in roots 100-fold more than in shoots of many species [34], mainly immobilized in root cell vacuoles [37]. Zinc in plant roots was significantly correlated with soil concentrations ($r = 0.75$ for its total and EDTA extractable fractions). Lead in roots and in the whole plant correlated with the total content in soil ($r = 0.70$ and 0.84 for root and total plant respectively), thus confirming the low translocation capacity from roots to shoots of this element. In most species it accumulated in roots [33], mainly localized in the insoluble fraction of cell walls, which is linked with the detoxification mechanism [38]. Instead, Cd content in all the plant organs resulted in being significantly correlated with soil PTE concentrations ($r = 0.89$, 0.87 and 0.83 respectively for its total, EDTA and $NH_4NO_3$ extractable fractions), therefore, confirming its highest mobility and the consequent well known health risk due to its entrance into the food chain [39,40].

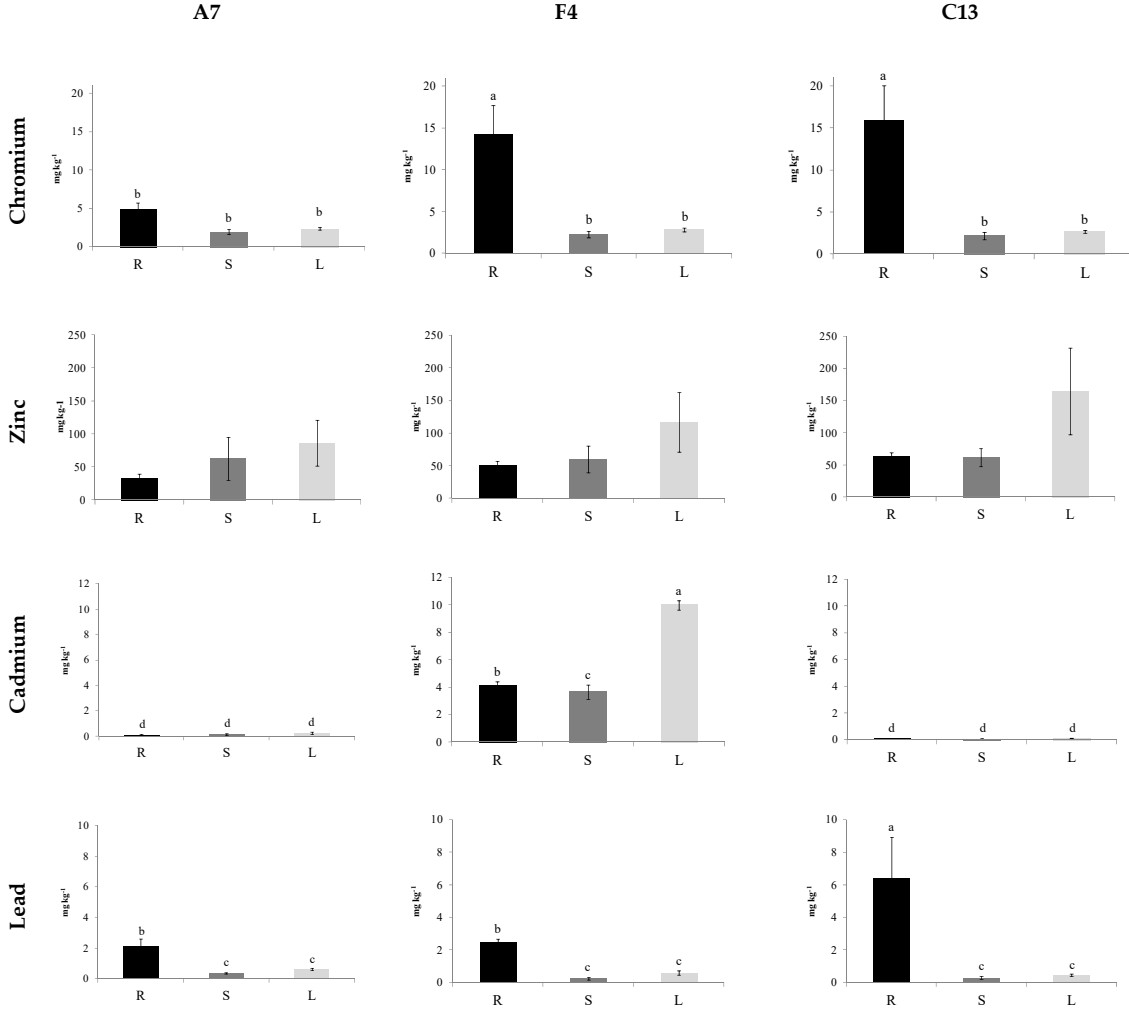

**Figure 1.** PTEs concentrations (mg kg⁻¹ dry weight) in roots (R), shoots (S) and leaves (L) of Indian mustard cultivated in mesocosms on the three studied soils (average of the three repeated croppings). (different letters indicate significant differences between the plant tissues in a given soil, for $p < 0.05$). Error bars represent standard deviations.

### 3.3. Changes in Soil Properties and PTE Bioavailability

Figure 2 shows the pH value and organic carbon content of soils A7, C13 and F4 measured just after the field sampling (T0) and in mesocosms treated with three croppings of Indian mustard (T1, T2, T3). The initial pH was neutral and very similar in the three soils (between 7.0 and 7.1; see Section 3.1. Soil Properties). Across the three croppings, a significant alkalization occurred in all soils up to values of 7.8 ± 0.1, 7.9 ± 0.1, 8.0 ± 0.1 in A7, C13, and F4, respectively. Exceptions to this trend were observed after the first cycle of *Brassica juncea* (T1) in A7 and F4, where the pH decreased respectively to 6.6 ± 0.3 and 7.0 ± 0.2 (although not significantly in F4). By contrast, the organic carbon content of the three soils initially was very different (between 13.1 and 25.9 g kg⁻¹; see Section 3.1. Soil Properties). At the end of the three croppings, a significant decline occurred in soils C13 and F4 down to values of 22.8 ± 0.9 and 14.6 ± 2.0 g kg⁻¹, respectively. A slight significant increase after the first cycle of *B. juncea* was observed in A7 up to values of 14.5 ± 0.9 g kg⁻¹ (Figure 2), while in C13 and F4 the content of organic carbon remained constant along the first two cropping cycles. When plants grow in soil with high concentration of heavy metals, changes of pH and organic carbon concentrations may occur (mainly in the rhizosphere), influencing the metal solubility and their chemistry at the root:soil interface [41]. The soil pH changes have been related to the balance of cation or anion uptake by plants and concomitant release of H⁺/OH⁻ and organic acids [42]. Studies on the effect of *Brassica juncea* roots

on rhizosphere chemistry detected a significant increase in soil solution pH and specific exudation pattern, resulting in significant changes to soil solution metals in the rhizosphere [43].

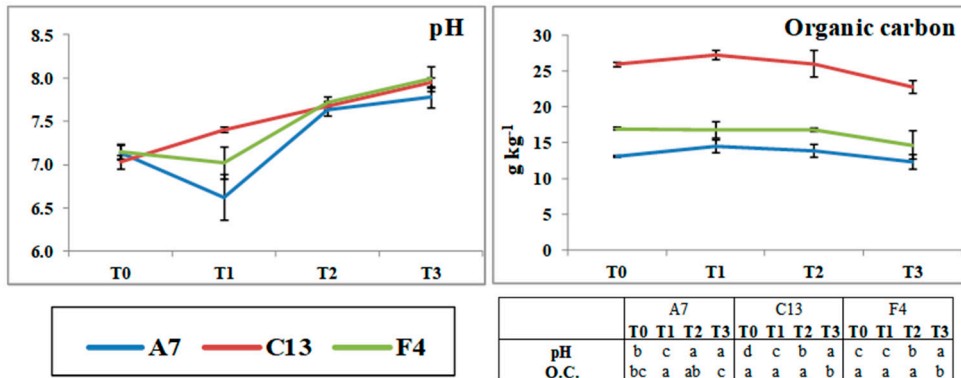

**Figure 2.** pH and organic carbon (O.C.) content in the three soils at T0 and after each cycle of *Brassica juncea* (T1, T2, T3). Different letters for each variable in the table indicate significant differences ($p < 0.05$) between the analysis times for each soil. Error bars represent standard deviations.

Figure 3 and Table S6 present the EDTA- and $NH_4NO_3$-extractable Cr, Zn, Cd, and Pb concentrations measured at the different steps of the mesocosm experiment. Overall, the potentially bioavailable amounts (as assessed by EDTA extraction) of the considered PTEs significantly decreased in soil after the successive croppings of *Brassica juncea*. This trend had few significant exceptions. The potentially bioavailable amounts of Cr in C13 increased after the first (T1) and second cropping cycle (T2) to decline to levels lower than T0 after the third cycle (T3). Cadmium in F4 remained constant after T1, then increased at T2, and the level did not change at T3.

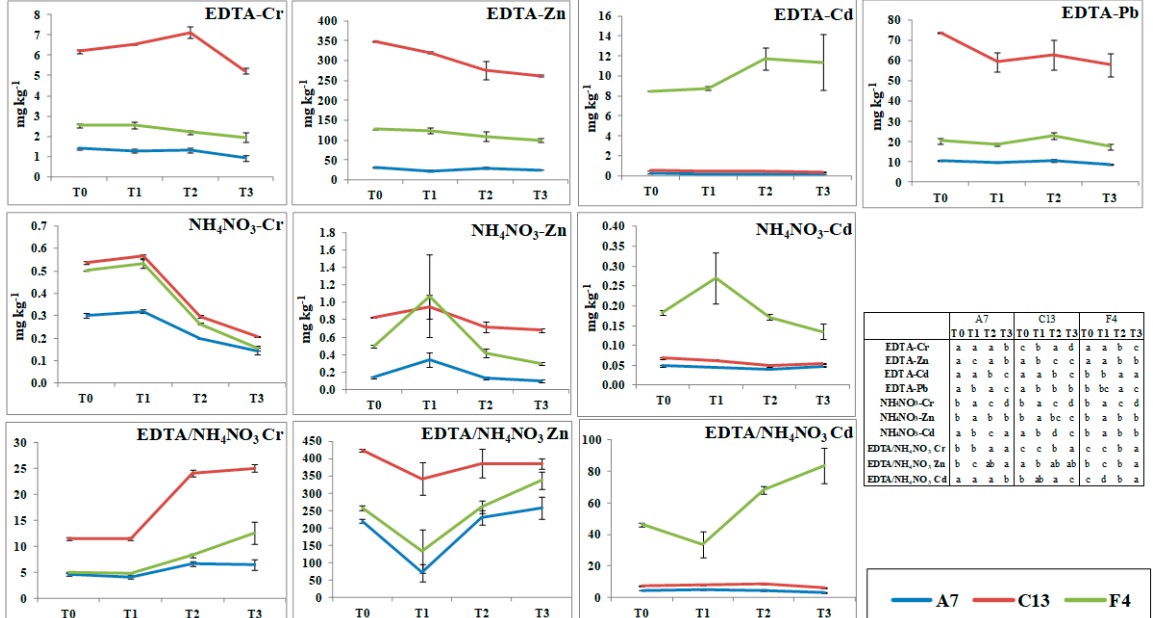

**Figure 3.** Amounts of Cr, Zn, Cd, and Pb extracted in EDTA and in $NH_4NO_3$, and ratio EDTA/$NH_4NO_3$ in the three studied soils at T0 and after each cycle of *Brassica juncea* (T1, T2, T3). Different letters for each variable in the table indicate significant differences ($p < 0.05$) among the times for each soil. Error bars represent standard deviations.

With respect to the initial amount (T0), at T3, the potentially bioavailable amounts for Cr declined to average values of 0.94, 2.0, and 5.2 mg kg$^{-1}$, corresponding to a total variation of −34, −23 and −16%

in A7, F4 and C13 respectively; for Zn to average values of 25.1, 100 and 262 mg kg$^{-1}$, corresponding to a total variation of −19, −21, −25% in A7, F4 and C13; for Pb to average values of 8.7, 17.5 and 57.9 mg kg$^{-1}$, corresponding to a total variation of −18, −14, −21% in A7, F4 and C13; for Cd to averages values of 0.15 and 0.34 mg kg$^{-1}$, corresponding to a total variation of −34 and −33% in A7 and C13. In F4, Cd potentially bioavailable amounts at T3 increased up to 11.4 mg kg$^{-1}$, corresponding to a positive variation of +34%.

The readily bioavailable amounts (as assessed by $NH_4NO_3$ extraction) generally increased significantly at T1 and decreased subsequently at T2 and T3 with more marked decrements between T1 and T2 than between T2 and T3 (Figure 3). The only exception to this trend was Cd, in which A7 and C13 decreased at T1 and T2 and increased at T3. The general increase at T1 was particularly marked for Zn in A7 and F4 (+141 and +118%, Table S6), while it was small for Cr (+6.1% in A7 and F4, and +4.9% in C13, Table S6). The total reduction at T3 with respect to T0, for Cr, amounted to −53% (up to 0.14 mg kg$^{-1}$) in A7, −69% (up to 0.16 mg kg$^{-1}$) in F4, and −61% (up to 0.21 mg kg$^{-1}$) in C13; for Zn to −30% (up to 0.10 mg kg$^{-1}$) in A7, −39% (up to 0.30 mg kg$^{-1}$) in F4, and −17% (up to 0.68 mg kg$^{-1}$) in C13; for Cd to −2.8% (up to 0.047 mg kg$^{-1}$) in A7, −26% (up to 0.14 mg kg$^{-1}$) in F4, and −20% (up to 0.055 mg kg$^{-1}$) in C13 (Figure 3; Table S6). In soil F4, Cd concentrations were significantly reduced after repeated croppings of *Brassica juncea* (from 0.18 mg kg$^{-1}$ at T0 to 0.14 mg kg$^{-1}$ at T3), but not below the trigger value of 0.1 mg kg$^{-1}$ set for Cd by some European countries for the pollutant transition soil-food plant on agricultural areas [21].

Summarizing, the repeated croppings of *Brassica juncea* in the three studied soils determined the reduction of both potentially and readily bioavailable contents (as assessed by EDTA and $NH_4NO_3$ extractions) of the considered PTEs. The only exception to this trend was shown by the increase of potentially bioavailable Cd in the F4 soil containing the element in amounts two and a half times higher than the threshold for agricultural soils (5 mg kg$^{-1}$ Cd). However, if we take into consideration the ratio between EDTA and $NH_4NO_3$ extracted amounts, for all the elements in the three soils it shows a general increase across the three croppings of *Brassica juncea* (Figure 3). For Cr, the ratio ranged at T0 from 4.7 in A7 to 11 in C13; it remained constant from T0 to T1 and increased from T1 to T2 and T3, where it reached a value of 25 in C13; for Zn the ratio at T0, it ranged from 220 in A7 to 424 in C13, it decreased from T0 to T1, and increased from T1 to T2, and T3 reaching a level slightly higher or similar to T0; for Cd, the ratio at T0 ranged from 4.6 in A7 to 46 in F4, it decreased in F4 from T0 to T1 and increased along T2 and T3 reaching the value of 84, whereas in A7 and C13 it remained almost constant in T1 and T2 and slightly decreased at T3. These results might be indicators of a process of re-equilibration between the soil phases taking part to the bioavailable pools toward a reduced relative mobility of the studied elements. This process did not occur after the first *Brassica juncea* cycle but started only during the second cropping.

The changes in PTEs mobility and bioavailability might be the combined result of the observed variation in pH and Organic Carbon (O.C.) content induced by plant growing, and the plant uptake. Indeed, significant correlations were found between the PTEs bioavailable amounts (extracted in EDTA and $NH_4NO_3$) and the soil properties (pH and organic carbon) measured at T0 and after each cropping cycle of Indian mustard in each soil (Table S7). In general and as expected, pH was always negatively and in most of the cases highly significantly correlated with amounts of PTEs extracted by both EDTA and $NH_4NO_3$. The Pearson correlation ($r$) in EDTA extract was −0.84 for Cr (in F4), $\geq$−0.77 for Zn (for C13 and F4), $\geq$0.74 for Cd (in all soils) and $r$ = −0.66 for Pb (in C13). While the $NH_4NO_3$ extracts showed a $r \geq$ −0.89 and −0.54 for Cr and Zn respectively in all soils, and for Cd only in C13 and F4 ($r \geq$ −0.76). Whereas, an opposite trend characterized organic carbon, with a significant correlation in EDTA extract only for Cr, Zn and Cd ($r \geq$ 0.53, see Table S7), and only for Cr ($r \geq$ 0.54) and Zn ($r$ = 0.56) for $NH_4NO_3$ extract. This may explain the general increase in the quantities of PTEs extracted in $NH_4NO_3$ at T1, when soil acidification occurred (paired with a general decrease of the ratio EDTA/$NH_4NO_3$), as well as the general reduction in these quantities at T2 and T3, when soil alkalinization takes place (paired with a general increase of the ratio EDTA/$NH_4NO_3$).

### 3.4. Bioavailability and Health Risk Assessed by Rocket Salad Growth and Metal Uptake

The evaluation of health risk deriving from the use of Rocket salad as bioassay is shown in Table 3 where the concentrations of analyzed PTEs in plant leaves (i.e., the plant edible parts) are reported. Only Rocket salad plants cultivated on F4 cadmium contaminated soil show Cd concentrations in leaves higher than the European legal threshold (EC Reg. 1881/06: 0.20 mg Cd kg$^{-1}$ fresh weight). Repeated phytoextraction with Indian mustard reduced Cd concentration in Rocked salad from the average value of 1.01 mg kg$^{-1}$ in the 1st year to 0.79 in the 3rd year. The health risk assessment by the Hazard Quotient approach did not highlight any risk associated to dietary exposure to Cr, Zn and Pb (HQ very close to 0) in spite of the very high concentrations of these PTEs in soil. In contrast, the HQ for Cd, although below 1, represented around 50% of the total Cd intake associated to potential health risks.

**Table 3.** PTE concentrations in leaves of Rocket Salad (mg kg$^{-1}$ fresh weight) at the first (I) and fourth (VI) harvest, and HQ estimation.

| Soil | Year | Harvest | Chromium | | | Zinc | | | Cadmium | | | Lead | | |
|------|------|---------|----------|------|------|-------|-------|------|----------|----------|------|------|----------|------|
| | | | Mean | S.E.* | HQ | Mean | S.E.* | HQ | Mean | S.E.* | HQ | Mean | S.E.* | HQ |
| **A7** | 2017 | I | 0.22 | 0.01 | 0.00 | 8.08 | 0.66 | 0.01 | 0.03 | 0.00 | 0.02 | 0.05 | 0.00 | 0.00 |
| | | IV | 0.45 | 0.06 | 0.00 | 6.95 | 0.79 | 0.01 | 0.03 | 0.00 | 0.02 | 0.17 | 0.02 | 0.01 |
| | 2018 | I | 0.18 | 0.02 | 0.00 | 6.14 | 0.61 | 0.01 | 0.02 | 0.00 | 0.01 | 0.05 | 0.01 | 0.00 |
| | | IV | 0.36 | 0.04 | 0.00 | 5.79 | 0.86 | 0.01 | 0.02 | 0.00 | 0.01 | 0.09 | 0.01 | 0.01 |
| | 2019 | I | 0.34 | 0.05 | 0.00 | 5.17 | 0.36 | 0.01 | 0.02 | 0.00 | 0.01 | 0.05 | 0.01 | 0.00 |
| | | IV | 0.25 | 0.01 | 0.00 | 4.24 | 0.13 | 0.00 | 0.02 | 0.00 | 0.01 | 0.06 | 0.01 | 0.00 |
| **F4** | 2017 | I | 0.26 | 0.02 | 0.00 | 15.88 | 1.75 | 0.02 | **0.85** | 0.09 | 0.50 | 0.08 | 0.00 | 0.01 |
| | | IV | 0.76 | 0.12 | 0.00 | 12.50 | 1.23 | 0.01 | **1.16** | 0.13 | 0.69 | 0.21 | 0.04 | 0.02 |
| | 2018 | I | 0.29 | 0.04 | 0.00 | 12.99 | 1.98 | 0.01 | **0.97** | 0.16 | 0.58 | 0.05 | 0.01 | 0.00 |
| | | IV | 0.36 | 0.07 | 0.00 | 8.35 | 0.87 | 0.01 | **0.65** | 0.04 | 0.38 | 0.06 | 0.01 | 0.01 |
| | 2019 | I | 0.61 | 0.07 | 0.00 | 8.91 | 0.41 | 0.01 | **0.81** | 0.08 | 0.48 | 0.05 | 0.00 | 0.00 |
| | | IV | 0.29 | 0.03 | 0.00 | 6.91 | 0.33 | 0.01 | **0.78** | 0.04 | 0.46 | 0.04 | 0.00 | 0.00 |
| **C13** | 2017 | I | 0.31 | 0.02 | 0.00 | 17.45 | 0.42 | 0.02 | 0.03 | 0.01 | 0.02 | 0.05 | 0.00 | 0.00 |
| | | IV | 0.63 | 0.02 | 0.00 | 21.11 | 0.41 | 0.02 | 0.03 | 0.00 | 0.02 | 0.16 | 0.02 | 0.01 |
| | 2018 | I | 0.22 | 0.02 | 0.00 | 16.73 | 1.73 | 0.02 | 0.02 | 0.00 | 0.01 | 0.04 | 0.01 | 0.00 |
| | | IV | 0.44 | 0.07 | 0.00 | 13.87 | 1.60 | 0.01 | 0.02 | 0.00 | 0.01 | 0.08 | 0.02 | 0.01 |
| | 2019 | I | 0.41 | 0.04 | 0.00 | 13.27 | 1.24 | 0.01 | 0.02 | 0.00 | 0.01 | 0.04 | 0.01 | 0.00 |
| | | IV | 0.33 | 0.03 | 0.00 | 12.66 | 1.01 | 0.01 | 0.02 | 0.00 | 0.01 | 0.05 | 0.01 | 0.00 |

* S.E. = Standard Error; Bold type is used to mark the values exceeding thresholds of EC Reg. 1881/06.

### 3.5. Phytoextraction Efficiency and Time-Span of Soil Phytoremediation

Total element extraction (TEE) for each PTE of interest obtained for the three soils by the successive croppings of Indian mustard was calculated according to Equation (1) given in Keller and Hammer [9] and was referred to a unit of soil:

$$\text{TEE} = \left( \sum_{i=1}^{3 \text{ crop}} \left( [\text{Element}]_{\text{plant}} \times \text{Weight}_{\text{plant}} \right) \right) \div \text{kg}_{\text{mesocosm\_soil}} \tag{1}$$

The total Cr extracted was very low for the three soils (respectively, 0.20 ± 0.03, 0.37 ± 0.09 and 0.45 ± 0.10 mg kg$^{-1}$ dry soil for soil A7, F4 and C13). For Zn, TEE was 5.0 ± 1.6, 6.1 ± 1.7 and 7.8 ± 1.5 mg Zn kg$^{-1}$ dry soil respectively in soils A7, F4 and C13. For Cd, the TEE reached 0.46 ± 0.09 mg kg$^{-1}$ dry soil for soil F4, while values were equal or below 0.01 mg kg$^{-1}$ for soils A7 and C13, respectively. For Pb, TEE was also very low (0.05 ± 0.01 in soil A7, 0.06 ± 0.01 in soil F4 and 0.11 ± 0.01 in soil C13 mg Pb kg$^{-1}$ dry soil).

These total amounts of elements extracted by Indian mustard across the three croppings were compared with the bioavailable amounts measured at the beginning and the differences reached at the end of the experiment (see Figure 3, Table 1 and Table S6). For Cd, the total amount removed

by Indian mustard from F4 soil was two and a half times larger than the amount extracted by $NH_4NO_3$ at T0 (0.46 vs. 0.18 mg kg$^{-1}$ dry soil) and 10 times larger than the difference measured in the $NH_4NO_3$-extractable between T0 and T3 (0.46 vs. 0.048 mg kg$^{-1}$ dry soil). The fact that the total exported Cd amount was larger than the $NH_4NO_3$-bioavailable amount measured at the beginning of the experiment can be explained by a large buffering capacity of the soil that replenished the readily bioavailable pool and therefore supplied Cd at a rate that was nearly as fast as the Cd uptake by plants. Such a replenishment was already observed by Keller and Hammer [9] for Cd and Zn after repeated croppings of *Thlaspi caerulescens* in metal contaminated soils. Taking into account that the amounts of Cd extracted by EDTA and the ratio of EDTA/$NH_4NO_3$ (Figure 3) significantly increased across the three croppings; it seems that with the plant uptake, a re-equilibration process among Cd forms in soil takes place. This process seems to produce an increase of the potentially bioavailable Cd pool, increasing the ability of the soil to buffer the changes induced by plant uptake across the repeated croppings. This process would guarantee a significant metal extraction over time until a stable new equilibrium will be reached for the Cd forms in soil and a new cropping does not extract as much Cd as the previous ones; on the other hand, the traces of this trend are already evident from the decrease in plant concentrations and in total amount of Cd extracted (both in Indian mustard and rocket salad) over time (Figure 4, Table S3, Tables 2 and 3).

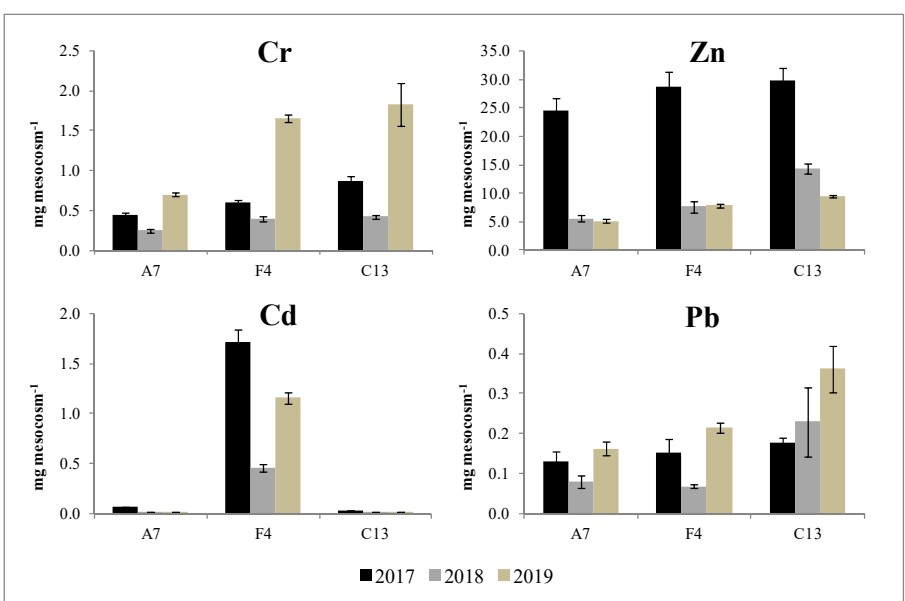

**Figure 4.** PTE uptake by Indian mustard (mg d.w. mesocosm$^{-1}$). The corresponding analysis of variance is given in Table S3. Error bars represent standard deviations.

Also for Zn, the total amounts removed by Indian mustard from F4 and C13 potentially polluted soils were far larger than the amounts extracted by $NH_4NO_3$ at T0 (6.1 and 7.8 vs. 0.49 and 0.82 mg kg$^{-1}$ dry soil) and the differences measured in the $NH_4NO_3$-extractable between T0 and T3 (6.1 and 7.8 vs. 0.19 and 0.14 mg kg$^{-1}$ dry soil). However, in contrast with what is observed for Cd, in both soils the EDTA-extractable amounts of Zn were significantly lower after the third cropping (from 127 to 100 mg kg$^{-1}$ in F4 and from 349 to 262 mg kg$^{-1}$ in C13). This might suggest that the buffering capacity of the soils replenished the Zn uptaken by plants, but that such a replenishment was progressively eroding the bioavailable pool. Also for Zn, the plant concentration and uptake are in accordance with this ri-equilibration trend (Figure 4, Table S3, Tables 2 and 3).

For Cr, the total amounts removed by Indian mustard from F4 and C13 potentially polluted soils were lower than the amounts extracted by $NH_4NO_3$ at T0 (0.37 and 0.45 vs. 0.50 and 0.54 mg kg$^{-1}$ dry soil) and equal to the differences measured in the $NH_4NO_3$-extractable between T0 and T3 (0.37 and 0.45 vs. 0.35 and 0.33 mg kg$^{-1}$ dry soil). Moreover, in both soils the EDTA-extractable amounts of

Cr were very low (<1 percent of the total) and remained almost constant after the three croppings (from 2.5 to 1.9 mg kg$^{-1}$ in F4 and from 6.2 to 5.2 mg kg$^{-1}$ in C13).

For Pb, the total amounts removed by Indian mustard from studied soils and the amounts extracted by $NH_4NO_3$ were negligible.

The very low metal availability might have been responsible for the limited Cr and Pb phytoextraction by Indian mustard as well as the limited metal changes in soil.

At this point, we estimated the effect of phytoremediation by Indian mustard growth on studied soils in terms of number of growing cycles needed to reclaim the soil (i.e., the number of Indian mustard growing cycles to reduce Cd concentration in Rocket salad below the European threshold of 0.20 mg Cd kg$^{-1}$ fresh weight.). This calculation was done considering the decreasing trend of Cd concentration in rocket salad along the 3 years of phytoextraction and assuming linearity of the trend.

By solving the equation in Figure 5 with y = 0.20 mg kg$^{-1}$, the number of Indian mustard cycles needed to reduce Cd concentration in edible leafy vegetables below the European threshold was 8.4.

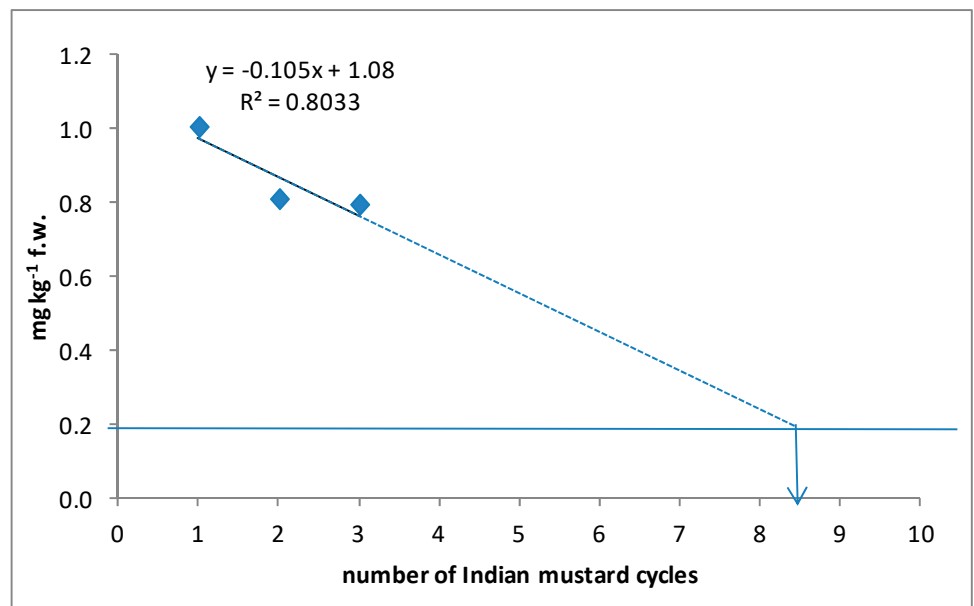

**Figure 5.** Projection of Cd concentration in Rocket salad (on the average of the year) in relation to the number of phytoextraction cycles by Indian mustard.

## 4. Conclusions

The results of this work show that three consecutive croppings of *Brassica juncea* on polluted soils, from an agricultural area under sequestration for illegal dumping of industrial sludge, can achieve a significant reduction of the bioavailable metal concentrations in soil along with changes of the soil pH and equilibrium between PTEs pools.

In the studied soils, PTE bioavailability was negatively correlated with pH and, according to single chemical extractions, Cd and Zn were more mobile and bioavailable than Cr and Pb. The different bioavailability was reflected by the PTEs uptake of Indian mustard plants extracting from soil and translocating from roots to shoots higher amounts of Zn and Cd than of Cr and Pb.

PTEs extraction by repeated croppings of Indian mustard changed across experimental years as a result of associated modification of soil properties due to the plant growth. This observation highlights that bioavailability in soil-plant systems is a dynamic concept changing with soil, plant and time. Legislations start to consider a reduction of bioavailable metal concentrations as an important step in soil remediation programs, providing valuable indications of the phytoextraction techniques efficiency. In this context, it is hardly surprising that we can predict PTE phytoavailability assuming

a static equilibrium of the soil system. Reactions in the soil environment are rarely at equilibrium, but instead are in a state of continuous change because of the dynamic processes occurring.

Of the studied PTEs, only cadmium was detected in the edible leaves of the bioindicator Rocket salad in concentrations exceeding the screening value for Cd given in the European regulation for food safety. The Hazard Quotient (ratio between average daily dose of Cd and its reference dose) confirmed the risks for human health due to consumption of Rocket salad, evidencing the efficacy of repeated croppings of Indian mustard for phytoremediation of the Cd contaminated soil. The decreasing trend of dietary exposure to Cd after repeated phytoextraction cycles allowed a rough but realistic assessment of the number of Indian mustard croppings required to reduce Cd levels in Rocket salad below the food safety threshold.

The phytoremediation approach represents an economically more realistic and cost-effective option than excavation, soil washing and in situ or off site soil disposal or even soil sealing, especially for indispensable agricultural areas producing food for the increasing world population. As observed in our study, it is possible and easier to phytoextract bioavailable instead than total Cd to clean up the soil. Such a reduction corresponds to a reduction of Cd concentration in Rocket salad, the plant taken as bioindicator. Assuming that this process is stable over time and can be modeled by a linear equation, the time-span necessary to take Cd concentration in Rocket salad below the European threshold of 0.20 mg kg$^{-1}$ f.w. was predicted. Nevertheless, across phytoextraction cycles, changes in soil pH, organic carbon and equilibrium between Cd pools of different bioavailability occur. In order to apply phytoextraction safely, these changes and the possible regeneration of the bioavailable pool have to be carefully monitored also when the complete phytoremediation of contaminated soil has been achieved. This in order to not overlook potential hazards stemming from further development of the vegetation, which might result in a change in metal distribution, mobility and availability in soil. In addition, with time an enhanced accumulation of organic matter by revegetation may give cause for concern and should also be monitored. The results support further investigation on the possibility to predict changes in bioavailability in the long term.

**Supplementary Materials:** The following are available online at http://www.mdpi.com/2073-4395/10/6/880/s1, Table S1: Parameters used for hazard quotient evaluation of Rocket salad (for adult people), Table S2: Analysis of variance of biomass yield expressed in g dry weight (d.w.) mesocosm$^{-1}$ and PTE concentrations (mg kg$^{-1}$ d.w.) in Indian mustard, Table S3: Analysis of variance of plant uptake of PTEs (mg mesocosm$^{-1}$), Table S4: Analysis of variance of PTE concentrations (mg kg$^{-1}$ d.w.) in plants grown in the three soils, Table S5: Correlation coefficients between soil (total, EDTA and $NH_4NO_3$) and plant (roots, shoots, leaves, total) concentrations of PTEs (mg kg$^{-1}$ d.w.), Table S6: Variation of soil PTE (extracted in EDTA and $NH_4NO_3$) after each Indian mustard cycle respect to the amount measured in the previous cycle (vs. T0 in brackets), Table S7: Pearson correlation coefficients between soil properties (pH and organic carbon) and amounts of Cr, Zn, Pb and Cd extracted in EDTA and in $NH_4NO_3$. O.C. = organic carbon.

**Author Contributions:** In this research, all authors contributed effectively. Conceptualization, M.F., P.A.; methodology, D.A., L.G.D., N.F., M.F., P.A.; formal analysis, D.A., L.G.D., E.C.; investigation, D.A., L.G.D., N.F., E.C.; resources, M.F., P.A.; writing—original draft preparation, D.A., L.G.D., M.F., P.A.; writing—review and editing, M.F., P.A.; visualization, D.A., L.G.D.; supervision, M.F., P.A.; project administration, M.F.; funding acquisition, M.F. All authors have read and agreed to the published version of the manuscript.

**Funding:** This research was funded by EC LIFE11/ENV/IT 275 Ecoremed and Italian MIUR PRIN2017BHH84R.

**Acknowledgments:** This study was carried out within the EC Project LIFE11 ENV/IT/000275 (ECOREMED). Special thanks go to Mario de Biase, Italian commissioner for remediation in Campania Region, who made available the study site for research activities.

**Conflicts of Interest:** The authors declare no conflict of interest. All authors read and approved the final manuscript.

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
