# Peer review of "Potentially Toxic Element Availability and Risk Assessment of Cadmium Dietary Exposure after Repeated Croppings of Brassica juncea in a Contaminated Agricultural Soil"

_agronomy, doi:10.3390/agronomy10060880_

Round 1

Reviewer 1 Report

See attachment

Author Response

Three soil lots from an area variably contaminated with Cd Cr Pb Zn by industrial effluents were potted and planted to Brassica juncea as a heavy metal extracting pre-crop, followed by Rocket salad as a model food, outdoors over 3 consecutive years. Accompanied by changes in soil pH and soil organic carbon, B. juncea reduced the concentrations of EDTA and NH4NO3 extractable elements as the potentially and readily available mineral pools by plant uptake. Salad accumulated Cd in amounts that exceeded the readily available NH4NO3 extractable resources. This may be well interpreted as restoration of the bioavailable pool from other soil fractions (but also as incredibility of NH4NO3 to define the readily available pool correctly).           

This interesting paper is well organized and convinces with the description of the test arrangements, the thorough discussion of the results, the optical presentations, and the considerations about restorations of the bioavailable mineral pool from other soil fractions.

Nevertheless, it is this latter aspect that misled to the conclusion that 8.4 Brassica crops turn contaminated soils, by depleting soluble metal fractions rather than their total stock, definitively into cropland. This evidence is lacking.    

We agree with the reviewer comment and we changed the text in conclusions accordingly (please see lines 529-533). Our sentence is based on the observed trend during the three years and it is attributed to the bioavailable fraction decrement which corresponds to a decreasing trend of Cd content in Rocket salad. This trend could allow to go below the European threshold in 8-9 Brassica cropping cycles. We will continue to monitor the mesocosms to increase the information and data for supporting our hypothesis.

14 delete one correspondence.

Done

17 … but plants can uptake only take up bioavailable forms…

Done

24  For Cd, this reduction did not bring the bioavailable amounts obtained by soil extraction with NH4NO3 below the trigger value of 0.1 mg kg-1 set by some European countries.

Done

45 …eco-friendly and cost-effective method with respect to the others [5]. Or in regard to the others.

Done

48 take up

Done

50 …concentrations of PTEs uptaking and accumulating them they accumulate in extremely high levels…

Done

55 …than the geogenic ones [5]. and this is also the reason of their dangerousness for the This may promote their transfer to other environmental compartments and to the food chain.

Done

57 Therefore, a realistic remediation objective through phytoextraction can be the progressive reduction of the contaminant to until safety levels of its the bioavailable portion of the contaminant, rather than 58 of the its total removal content [9].

Done

59 However, the bioavailability of a PTE is closely linked to the nature of the element, to the chemical forms in which it occurs in soil and to the chemical and physical characteristics of the soil itself. This determines the repartition of the total content of the element between the various soil geochemical fractions and the soil solution (in which the element is in a high readily bioavailable form) [3, 10].

Done

63 This repartition is controlled by dynamic equilibria between different forms elements and soil fractions. and this aspect It must be considered in planning strategies of phytoremediation. because After short-term phytoextraction (or only few cycles of phytoextraction), the decreased element in the readily bioavailable fraction of the respective element may be replenished through soil element re-equilibration, repartition between soil fractions and soil solution, and the kinetics of replenishedment can change over time [9].

Done

72  …were done to assess residual the bioavailability of the residual PTEs and thus the opportunity to reutilize such the treated soil for food production.

Done

85 an agricultural area?

We change “area” with “land”

86 …interested by past  contaminated by former illegal dumping 

Done

90 …in almost all regions. In contrast, Cd and …

Done

99 …which hinder depreciate comparisons between results…

Done

103 Soils F4 and C13 were identified respectively as Cd and Pb contaminated, respectively, on the basis of total contents. Soil A7 was collected from a sub-area in which no legal thresholds of PTEs were exceeding surpassed, was detected and it was considered as control.

Done

107 Three successive croppings of Indian mustard (Brassica juncea L. Czern.), alternated to sowing of followed by Rocket salad (Eruca vesicaria L. Cav.) were made following a similar process each year. established over three consecutive years.

Done

110 sown

Done

111 …the others uprooted removed.

Done

112 …the 5 plants per mesocosm were uprooted harvested and…

Done

114 …a portion was put in the oven ( dried at 60 °C until to constant weight) to determine…

Done

116 …content. at the moment of uprooting During harvest, the rhizosphere soil (i.e. the soil adhering to the roots) was also sampled from each mesocosm for the chemical analyses.

Done

118 sown

Done

119 …a normal productive process. cutting the leaves At commercial maturity, the leaves were cut four times; and analyzing the PTE content was only analyzed in the edible parts only at the end of the first and fourth cutting.

Done

135, 140 Why pseudo-total?

Because the soils were digested in aqua regia (as normally used for environmental investigation), and the term “pseudo-total” usually is referred to the concentration in soils obtained with this extraction that not mineralize all the silicate. (see Taraškevičius et al., 2012. Case study of the relationship between aqua regia and real total contents of harmful trace elements in some European soils. Journal of Chemistry, 2013).

148 Indicate the mode of “acid digestion” of plant tissues and, potentially, the use of microwave.

Done

163 Indicate provider of the software.

Done

180 occurring

Done

205 bind

Done

198 The investigated elements decrease their solubility at alkaline pH, and this is not the case in the three studied soils with neutral pH. 

How can you state that? You did not work with pH modified extractants. But see line 375 and 476.

We have modified the sentence to eliminate the misunderstanding

210 …extractability of cadmium in respect to the other…

Done

210-216 Please subdivide this cascade in single, short, and comprehensible sentences.

Done

219 A higher ratio can indicates the poor mobility or the tendency of PTEs to switch…

Done

265 …other than to the element’s low…

Done

273 …organs resulted in significantce (Table 273 S3).

Done

286 …the very low bioavailability bioconstruction capacity of acceptance by plants for this element [28]. that resulted It accumulated in roots 100-fold more than in shoots of many species [34], mainly immobilized in root cell vacuoles [37].

Done

288 Zinc in plant roots was significantly correlated with soil concentrations (r=0.75 for its total and EDTA extractable fractio). Lead in roots and in the whole plant resulted correlated with the total content in soil, thus confirming the low translocation capacity from roots to shoots of this element. so that In most species it resulted accumulated in roots [33], mainly localized in the insoluble fraction of cell walls, which is linked with the detoxification mechanism [38].

Done

294 entrance into the food chain.

Done

Table 3: Indicate ± values as SD?

We have added a specification.

Figures 1-4: Error bars ± SD?

We have added a specification.

Table 4: What is 8 d.f.?

d.f. stands for “degrees of freedom”, we have added a specification.

405 Table 6: What mean the roman numbers I and IV? Are they quadruplicate pots?

Indicate the number of the rocket salad harvest (first and fourth harvest). We have added a specification.

Reviewer 2 Report

Diana Agrelli et al. report on phytoextraction of Cr, Zn , Cd and Pb in an open-air mesocosm study with soil from 3 areas x 4 replicates.  The amount of soil per mesocosm was 7 kg.  Three successive crops of Brassica juncea L. Czern. alternated with crops of Eruca vesicaria L. Cav. as the food-plant bioindicator. The Rocket salad was cultivated simulating a normal productive process, cutting the mature leaves at four times; and analyzing the PTE content in the edible parts at the end of the first and fourth harvest.  Dietary exposure and risk assessment were calculated.

The study appears to have been well-conducted, and the chemical analyses are suitable; however, there is insufficient information about the fertiliser inputs used to promote plant production.  Those inputs should be documented as they will have contributed to the processes.

My estimate is that the depth of soil in the microcosms was ~10 cm.  That is, the roots might be expected to have exploited the soil in the microcosm more thoroughly than they would in the field.  The likely impact should be discussed.

Graphical presentation of the key results would improve readability with more of the underpinning data being transferred to the supplementary section.  Lastly, the English needs minor editing for clarity of meaning.

Author Response

Diana Agrelli et al. report on phytoextraction of Cr, Zn , Cd and Pb in an open-air mesocosm study with soil from 3 areas x 4 replicates. The amount of soil per mesocosm was 7 kg. Three successive crops of Brassica juncea L. Czern. alternated with crops of Eruca vesicaria L. Cav. as the food-plant bioindicator. The Rocket salad was cultivated simulating a normal productive process, cutting the mature leaves at four times; and analyzing the PTE content in the edible parts at the end of the first and fourth harvest. Dietary exposure and risk assessment were calculated.

The study appears to have been well-conducted, and the chemical analyses are suitable; however,

there is insufficient information about the fertiliser inputs used to promote plant production. Those inputs should be documented as they will have contributed to the processes.

We have added the sentence about the fertilizer in 2.2 paragraph (please see lines: 135-142)

My estimate is that the depth of soil in the microcosms was ~10 cm. That is, the roots might be expected to have exploited the soil in the microcosm more thoroughly than they would in the field. The likely impact should be discussed.

At the end of 2.1 paragraph, we explain our awareness of the different behavior of the plants between mesocosm and full field. Indeed a handle next step could be set a field experiment to evaluate what has been observed in current mesocosm experiment

Graphical presentation of the key results would improve readability with more of the underpinning data being transferred to the supplementary section.

We moved to supplementary section tables 1, 4 and 5 and changed some sentences in the text (please see lines: 305-306, 311, 315-316, 401-406) and the tables numeration.

Lastly, the English needs minor editing for clarity of meaning.

Many sentences have been improved